# Roles of Tumor-Infiltrating Lymphocytes and Antitumor Immune Responses as Predictive and Prognostic Markers in Patients with Breast Cancer Receiving Neoadjuvant Chemotherapy

**DOI:** 10.3390/ijms26209959

**Published:** 2025-10-13

**Authors:** Ryungsa Kim, Takanori Kin, Koji Arihiro

**Affiliations:** 1Department of Breast Surgery, Hiroshima Mark Clinic, 1-4-3F, 2-Chome Ohte-machi, Naka-ku, Hiroshima 730-0051, Japan; 2Department of Breast and Endocrine Surgery, Osaka University Graduate School of Medicine, 2-15 Yamadaoka, Suita 565-0871, Japan; ymj5014266@gmail.com; 3Department of Anatomical Pathology, Hiroshima University Hospital, 1-2-3 Kasumi, Minami-ku, Hiroshima 734-8551, Japan; arihiro@hiroshima-u.ac.jp

**Keywords:** breast cancer, neoadjuvant chemotherapy, tumor-infiltrating lymphocyte, pathological complete response, immune response

## Abstract

Tumor-infiltrating lymphocytes (TILs) are thought to play important roles in tumor shrinkage and survival prolongation in patients with breast cancer receiving neoadjuvant chemotherapy (NAC). TILs are mononuclear immune cells such as lymphocytes and plasma cells, including CD4+ and CD8+ T cells, natural killer cells, B cells, macrophages, regulatory T cells (Tregs), and myeloid/dendritic cells. The pre-NAC presence of more T cells and fewer Tregs in biopsy samples of primary breast tumors is known to contribute to tumor shrinkage and prolonged survival. This review was conducted to elucidate these roles in patients with breast cancer treated with NAC. Publications selected for inclusion in this review were identified by a PubMed search for articles published in English, performed using the terms “breast cancer”, “neoadjuvant chemotherapy”, “tumor-infiltrating lymphocyte”, “pathological complete response”, and “immune response”. The search was completed in July 2024. The functional roles of TILs in the achievement of these outcomes may vary by tumor subtype; increases and decreases in TIL levels before and after NAC have been shown to have conflicting effects. Biomarkers have been reported to predict local responses in the tumor microenvironment (e.g., immune-related gene signatures) and systemic immune responses (e.g., neutrophil-to-lymphocyte and platelet-to-lymphocyte ratios). Immune gene signatures and immune cell infiltration do not appear to be universally associated with tumor response or outcome in patients with breast cancer treated with NAC. The functional roles of TILs in breast tumor response and breast cancer survival may vary by tumor subtype, and conflicting results for the same subtypes may be due to differences in NAC regimens, immune responses, tumor heterogeneity, sample size, and the technical methods used to evaluate TILs in tumor samples.

## 1. Introduction

Tumor sensitivity to anticancer drugs and the activation of antitumor immunity are important factors determining breast tumor shrinkage after neoadjuvant chemotherapy (NAC) [1,2]. In this context, tumor-infiltrating lymphocytes (TILs) are thought to play an important role in the induction of antitumor immunity and, thus, primary tumor shrinkage and survival prolongation [3,4,5]. TILs are generated with tumor immune recognition as a host defense mechanism, and their presence is thus associated with improved outcomes. However, this presence has diverse positive and negative associations with outcomes and treatment responses, potentially related to the lymphocyte subtype and location and the molecular subtype of cancer [6].

TILs consist of mononuclear immune cells such as lymphocytes and plasma cells, including cluster of differentiation (CD)4+ and CD8+ T cells, B cells, natural killer (NK) cells, regulatory T cells (Tregs), myeloid/dendritic cells, and macrophages (M1 and M2 subtypes) [7]. Pathologically, they are classified as intratumoral and stromal. Stromal tumor-infiltrating lymphocytes (sTILs) reside in the tumor stroma between cancer cell populations and do not interact directly with cancer cells, whereas intratumoral tumor-infiltrating lymphocytes (iTILs) reside in tumors and come into direct contact with cancer cells [8]. Although this proximity may make iTILs more biologically important than sTILs, paradoxical results regarding the roles of these cells in breast tumor shrinkage have been reported [9,10].

Higher pre-NAC TIL levels have been associated with higher pathological complete response (pCR) rates and longer survival than have lower levels in triple-negative (TN) and human epidermal growth factor receptor 2 (HER2)+ breast cancers but not in hormone receptor (HR)+ and HER2− (luminal) breast cancers [11]. However, other studies have yielded conflicting results regarding the association of high pre-NAC TIL levels with pCR in specific tumor subtypes [12,13,14]. Moreover, post-NAC TIL levels and the expression of immune response genes associated with pCR decrease with tumor shrinkage in most cases, but residual HER2-type lesions have higher TIL levels, and an aggressive phenotype associated with tumor progression [13,15]. This review was conducted to elucidate the functional roles of TILs in breast tumor shrinkage and survival based on the latest published data and to help clarify how TILs enhance the efficacy of NAC for breast cancer.

## 2. Therapeutic Effects of Anticancer Drug–Induced Antitumor Immunity

High pre-NAC CD8+ TIL levels are known to play a major role in the therapeutic response in patients with breast cancer [11]. Immune cell activation is mediated by anticancer drugs, some of which (e.g., anthracyclines) induce immunogenic cell death (ICD) [16]. ICD, which improves the low immunogenicity of breast and other cancer cells, is characterized by spatiotemporal surface rearrangements of danger molecules and the simultaneous occurrence of damage-associated molecular patterns (DAMPs), such as calreticulin (CRT), adenosine triphosphate (ATP), high mobility group box 1 (HMGB1), and annexin A1 (ANXA1) [16,17].

DAMPs are endogenous molecules that have housekeeping functions in unstressed cells, but act as danger signals sensed by the immune system upon exposure to cellular stress or injury [16,17]. They stimulate the adaptive immune system by binding to homologous receptors on innate immune cells, such as dendritic cells (DCs), leading to tumor antigen-specific CD8+ T cell-mediated immune responses that eliminate residual cancer cells and establish immune-resident memory T cells [18]. Upon ICD induction, large amounts of DAMPs are released to activate the ICD signaling pathway, promote DC maturation, activate cytotoxic T lymphocytes (CTLs), and enhance antitumor effects [19]. ICD occurs in association with autophagy, ferroptosis, pyroptosis, and necroptosis to promote antitumor immunity [20].

During antitumor immunity activation, dying cancer cells release CRT to the extreme membrane surface and release DAMPs. DCs are recruited and activated by the secretion of ATP, which binds to P_2_RY_2_ and P_2_RX_7_ receptors. Secreted extracellular ATP functions as a “find me” signal to APCs via ATP binding to P_2_RY_2_ receptors, inducing APC chemotaxis. ATP signals via P_2_RX_7_ receptors on DC surfaces, activates NOD-, LRR- and pyrin domain-containing protein 3 (NLRP3) inflammasomes and promotes interleukin 1β (IL-1β) production. IL-1β stimulates DC activation and antigen presentation [16,17]. ANXA1 is released and binds to formyl peptide receptor 1 for homing [16,17,18]. CRT binds to CD91 receptors on DCs to stimulate the phagocytosis of dying cancer cells. The release of HMGB1 stimulates DC recruitment by binding to the receptor for advanced glycation products and induces DC maturation via toll-like receptor 4 signaling [16,17,18]. Immature DCs are also activated and mature with uptake by tumor antigens (TAs) released from dead cells. Mature DCs migrate to lymph nodes, where they undergo cross-priming to induce the clonal expansion of IL-12- and type I interferon (IFN)-producing T cells, which exert a cytotoxic response and eradicate cancer cells [18]. TAs and tumor-associated antigens (TAAs) are processed by mature DCs and presented as major histocompatibility complex I molecules to CD8+ T cells, generating CTLs for tumor-specific immune responses, leading to tumor regression [21].

## 3. Modulation of the Tumor Microenvironment by Anticancer Drugs to Enhance the Therapeutic Effect

The tumor microenvironment is shifted to an immunosuppressive state that promotes cancer progression by myeloid-derived suppressor cells (MDSCs) induced by tumor-derived soluble factors (TDSFs), such as vascular endothelial growth factor and prostaglandin E_2_ [22]. MDSCs include immature DCs, monocytes, and neutrophils, which express Fas L, bind to Fas on CD8+ T cells, and induce apoptotic CD8+ T cells via the C-X-C chemokine ligand 1/C-X-C chemokine receptor 2 signaling pathway [23,24]. Other immunosuppressors are Tregs and tumor-associated (M2) macrophages, which, like MDSCs, release immunosuppressive cytokines such as IL-10 and transforming growth factor beta (TGF-β) and increase programmed death ligand 1 (PD-L1) expression in tumor cells, resulting in immune evasion from CTL attacks [24]. This immune escape leads to breast tumor growth and metastasis.

Anticancer drugs induce cell death and the release of TAs and TAAs, which results in tumor-specific immune activation [21]. In contrast to protumor cytokines such as IL-10 and TGF-β, abundant antitumor cytokines such as IL-12 and IFN-γ modulate the tumor microenvironment for immune activation [21]. However, responses differ among breast cancer subtypes. For example, a post-NAC decrease in PD-L1 expression was observed in TN breast cancer but not in non-subtype-specified (NS) breast cancer [25,26].

## 4. TILs and Immune Responses as Predictive and Prognostic Markers of Treatment Response

### 4.1. TN Breast Cancer

In TN breast cancer, higher pre-NAC TIL levels and more diffuse TIL distributions are associated with higher pCR rates and are strong predictors of pCR [27,28,29,30,31,32,33,34]. pCR is also associated with the apoptosis score [34], patient age < 40 years with high TIL levels [35], lymphocyte-predominant breast cancer (LPBC) relative to non-LPBC [36], low neutrophil-to-lymphocyte ratios (NLRs) [28], higher pre-NAC CD8/CD4 ratios [37], high degrees of T cell receptor clonality, PD-L1 positivity, and the spatial proximity of T cells to tumor cells [37]. This proximity, in combination with PD-L1 expression, is predictive of pCR [37]. In addition, patients with more TILs before NAC and PD-L1 positivity who received anticancer drugs and immune checkpoint inhibitors had higher pCR rates [27,29]. Treatment-induced changes in TIL levels also correlate with the metabolic response [38]. Systemic immune mediator levels correlate with high TIL levels in pCR [39]. The pre-NAC CD8/forkhead box P3 (FOXP3) ratio is higher in pCR cases and is the only independent predictor of pCR [40,41].

iTILs and sTILs correlate with and are predictors of pCR [9,10]. sTILs densities are associated with the residual cancer burden (RCB) status of 0/I, and homologous recombination deficiency (HRD) is associated with RCB 0/I status and pCR [42]. Post-NAC residual disease (RD) TIL levels correlate with the CD8+ T cell density [43]. The TIL and PD-L1 levels decrease after NAC in pCR [25], and higher post-NAC CD4 levels are associated with pCR [31].

In TN breast cancer, high TIL levels are associated with improved overall survival (OS) and disease-free survival (DFS) [11,12]. Correspondingly, high pre-NAC TIL levels and low NLRs are associated with improved OS, whereas PD-L1 and T cell immunoglobulin levels and mucin domain-3 positivity are associated with worse OS [25,28]. Higher TIL densities and diffuse distribution patterns correlate with improved relapse-free survival (RFS) and OS [32]. Low NLRs and high TIL levels have also been associated with improved breast cancer-specific survival (BCSS) [28]. pCR is associated with improved OS and DFS [44]. In addition, circulating cytokines with TILs predict improved survival [39]. High TIL levels predict good prognosis, and decreased TIL levels may be associated with local recurrence [30]. DFS rates are better for LPBC than for non-LPBC [36]. Higher pre-NAC CD3+, CD4+, CD8+, and FOXP3+ T cell levels and CD4/FOXP3 ratios are associated with longer DFS, and higher pre-NAC CD3+ T cell levels are additionally associated with longer OS [31]. When the pCR or RD is small, the platelet-to-lymphocyte ratio (PLR) is a better predictor of less distant recurrence or longer distant recurrence-free survival than TILs [45].

High post-NAC TIL levels predict poor RFS and OS, and TIL levels are associated negatively with event-free survival (EFS) [38,46]. In contrast, RD with high TIL levels after NAC was associated with prolonged OS and BCSS in a meta-analysis [47] and with better distant DFS in another study [48]. In RCB II cases, RD with high TIL levels is associated with improved RFS and OS [43]. These findings are summarized in Table 1 and Figure 1.

### 4.2. HER2+ Breast Cancer

In various studies on HER2+ breast cancer, TIL levels have shown significant association, a nonsignificant trend of association, and no association with pCR [13,50,51]. High TIL levels and CD8 infiltration have been identified as predictors of pCR [30,33,52,53,54,55,56]. In addition, TIL levels increased in non-pCR cases, while CD8+ T cell levels decreased in tumors under treatment showing pCR [57]. High TIL and PD-L1+ TIL levels are predictors of pCR, and PD-L1+ tumor cell levels are reduced with trastuzumab and chemotherapy [58]. High Ki-67 levels are also strong predictors of pCR, with TILs and FOXP3+ T cells potentially involved in the tumor response [59]. Multiple B cell-related signatures have also been associated with pCR in HER2+ breast cancer [60].

Immune cell (*n* = 23) activity clusters correlate with TIL levels and better predict pCR [61]. iTILs and sTILs correlate with pCR [10]. sTIL levels are associated with pCR and increased CD3+, CD3+ CD8− FOXP3−, CD8+, and FOXP3+ levels [62]. The pCR was found to be higher in cases with higher CD8/FOXP3 ratios [41]. Higher CD4+, CD8+, and CD20+ sTIL levels and CD20+ iTIL levels were associated with higher pCR rates [51]. In addition, an association between pCR and higher baseline infiltration by sCD4+, iCD4+, and iCD20+ TILs was observed [51]. However, increased pretreatment TIL, but not sTIL, levels are associated with pCR, whereas the opposite is true in tumors undergoing treatment [57]. In addition, neither iTILs nor sTILs are predictors of pCR [9]. The magnitude of the post-NAC decrease in TIL levels was associated with pCR, and higher TIL levels in RD after NAC was found to indicate an aggressive phenotype [63].

In HER2+ breast cancer, high pre-NAC TIL levels are associated with better 3-year invasive DFS, regardless of pCR [50], and, in general, with improved OS (see Denkert et al. [12]), DFS, and EFS [11,64]. They predict good prognosis, whereas decreased TIL levels may be associated with local recurrence [30]. In triple-positive breast cancer, TIL levels are more prognostically relevant than pCR [56]. In contrast, pCR is associated with improved DFS and OS in HER2+ breast cancer [44].

An inferred immune cell activity cluster was found to better predict DFS and OS [61]. Multiple B cell-related signatures were found to be associated more strongly with EFS than TIL levels were [60]. Increased post-NAC TIL levels have been reported to be associated with improved BCSS and DFS [54] and with impaired DFS [13]. Higher post-NAC TIL levels in RD were associated with poor outcomes [15,63]. These findings are summarized in Table 2 and Figure 2.

### 4.3. Luminal Breast Cancer

In luminal breast cancer, TIL levels, immune-related gene signatures, and specific immune cell subpopulations are associated with pCR [64]. Pre-NAC TIL levels were found to be higher in pCR cases [13,14], but a meta-analysis demonstrated that high TIL levels do not predict pCR [11]. In contrast, other studies have found no biomarkers that correlate with pCR [10]. The sequential administration of anthracyclines and anti-PD-1 agents may activate an antitumor immune response in the basal molecular subtype of luminal B breast cancer [65]. The expression of an immune-related gene signature is associated with higher pCR rates and reduced tumor size; with the TIL level, it is as predictor of tumor shrinkage [66]. pCR also correlates with intratumoral CD8+ TIL levels [67].

CD8+ TIL levels correlate with DFS and OS. Strong iCD8+ TIL expression is associated with improved OS and is an independent prognostic factor for OS [67]. Unlike for TN and HER2+ breast cancers, however, high TIL levels have been associated with worse survival of luminal breast cancer [11,12]. Low FOXP3 levels are associated with improved DFS [66]. In contrast, pre-NAC TIL > 10% abundance and lymphocyte-to-monocyte ratios ≤ 5.2 correlate with worse DFS and OS [68]. Low post-NAC TIL levels have been associated with better RFS [14]. These findings are summarized in Table 3 and Figure 3.

### 4.4. Non-Subtype-Specified (NS) Breast Cancer

In NS breast cancer, elevated thin protein expression has been associated with pCR [69]. CD8+ T cell infiltration, but not the CD4 or FOXP3 level, was associated with higher pCR rates [70]. Chemoattractant cytokines such as chemokine (C-C motif) ligand 21 (CCL21) and CCL19, as well as CTL markers, were associated with increased pCR rates, and stromal functions have been associated with RD [71]. Plasma cell and B cell infiltration correlate with pCR [72]. High levels of CD4+, CD8+, and FOXP3+ TILs were associated with pCR; this association held for CD8+ TIL levels regardless of the breast cancer subtype, stem cell phenotype, epithelial–mesenchymal transition, and treatment regimen [73]. PD-L1 and programmed death 1 (PD-1) expression correlate with increased TIL levels and pCR [74]. B cell/plasma cell, myeloid cell/DC, and T cell/NK cell metagenes have been associated with favorable pathological responses, with the former two having the most important roles [75].

High pre-NAC TIL levels, high PD-L1 levels, the NLR, and the PLR were found to be predictive markers of pCR [5,12,44,76,77,78,79,80]. The TIL level may also be a predictive marker of residual postoperative lymph-node involvement [80]. iTIL levels are predictors of pCR, whereas contradictory results have been obtained for sTIL levels [9]. The CD8/effector Treg (eTreg) ratio is an independent predictor of pCR [81]. Stromal auto-quantitative analysis (AQUA) scores for CD3, CD8, and CD20 predicted pCR, with the latter association holding regardless of clinicopathological factors [82].

After NAC, the reduction in the mean TIL level is associated with pCR [13]. In addition, the expression of most immune response genes is decreased in pCR cases [71]. Higher pre-NAC TIL levels and PD-L1 positivity have been associated with higher pCR rates and TIL levels, but PD-L1 expression has been found to be reduced after NAC [26].

Pre-NAC TIL levels are associated with DFS [5,13,81]. Similarly, high TIL levels have been associated with prolonged DFS and OS [70]. High CD8/FOXP3 ratios predict good prognosis [41], whereas PD-L1 and PD-1 expression correlate with poor prognosis [74]. Decreased post-NAC TIL levels in pCR cases have been identified as a surrogate marker of longer DFS [78]. Plasma cell infiltration correlates with the prolonged DFS of patients with HR-negative breast cancer [72]. These findings are summarized in Table 4 and Figure 4.

## 5. Overview of Predictive and Prognostic Factors Across Different Subtypes

In the TN subtype, TILs, low NLR/high TIL density, systemic immune mediators, high TIL levels/PD-L1 (chemotherapy + ICI), CD8/FOXP3 ratio, TIL level changes, high TCR clonality, PD-L1, high CD3/CD68 ratio, T cell–tumor cell proximity, T cell proximity, PD-L1, decreased TIL levels and PD-L1, high pre-NAC TIL levels, CD8/CD4 ratio, post-NAC CD4 levels, and high apoptosis scores were associated with pCR. Patients under 40 years of age, HRD, and LPBC were associated with pCR. T cell immunoglobulin mucin-3 (TIM-3) positivity was observed more frequently in non-pCR cases. Favorable prognosis was associated with high pre-NAC TIL levels; high pre-NAC TIL density/diffuse TIL pattern; low NLR/high TIL levels; circulating cytokines associated with TILs; high pre-NAC CD3, CD4, CD8, and FOXP3 levels; CD4/FOXP3 ratio; and high pre-NAC CD3 levels, PLR, and high post-NAC TIL levels in residual disease (RD). Poor prognosis was associated with decreased or increased post-NAC TIL levels, as well as PD-L1/TIM-3 positivity.

In the HER2 subtype, TIL levels; CD8+ infiltration; inferred immune cell activity; multiple B cell-related signatures; CD3+, CD3+CD8−FOXP3−, CD8+, and FOXP3+ sTILs; immune cell aggregation; PD-L1; high Ki-67 levels; high CD4+, CD8+, and CD20+ sTILs; CD20+ iTIL levels; and LPBC were associated with pCR. Although TIL levels decreased, high TIL levels after NAC in RD indicated progressive disease. Favorable prognosis was associated with high pre-NAC TIL levels, regardless of pCR status, inferred immune cell activity clusters, multiple B cell-related signatures, and increased post-NAC TIL. Poor prognosis was associated with high post-NAC TIL levels in RD, while decreased post-NAC TIL levels were associated with local recurrence.

In the luminal subtype, TIL levels, immune-related gene signatures, immune cell subpopulations, gene expression profiling of FOXP3+ T cells and CD163+ macrophages, and iCD8+ TIL levels were associated with pCR. Favorable prognosis was associated with low FOXP3 levels, CD8+ TIL levels, and iCD8+ TILs.

In the NS subtype, pre-NAC TIL levels were associated with pCR regardless of subtype. Protein titin, NLR, PLR, CD8/eTreg ratio, CD8/FOXP3, plasma cell/B cell infiltration, myeloid/DCs, T cell/NK cells, chemotactic cytokines (CCL21, CCL19), CTL markers, and PD-L1/PD-1 expression were associated with pCR. The mean TIL level decreased after NAC. In different subtypes, multiple conflicting results were observed between high TIL levels, iTIL and sTIL levels, and pCR. Favorable prognosis was associated with pre-NAC TIL levels, plasma cell infiltration in HR-negative tumors, high TIL levels in TN/HER2, increased TIL levels in TN (though OS was shortened in luminal type), low post-NAC TIL levels in luminal type, and high CD8/FOXP3 ratio. Poor prognosis in HER2+ tumor was associated with high post-NAC TIL levels and PD-L1/PD-1 expression.

As predictive factors, high pre-NAC TILs, CD8+ infiltration, and FOXP3 expression are pCR-associated factors common to all subtypes. PD-L1 expression and LPBC are overlapping factors observed in TN and HER2 subtypes, while B cell infiltration is an overlapping factor observed in HER2 and NS subtypes. Table 5 summarizes the predictive and prognostic factors for different breast cancer subtypes following NAC treatment.

## 6. Possible Causes of Conflicting Findings for TILs and Future Approaches

### 6.1. Correlation Between TIL, iTILs, sTILs and pCR and Prognosis

The reasons why significant and insignificant associations were observed between pre-NAC TIL and pCR in HER2+ breast cancer are as follows [13]: (i) differences in tumor biology; (ii) differences in the quantity and quality of immune cell infiltration and the thresholds corresponding to the definition of high TIL tumors; (iii) types of anti-HER2-targeted therapy and their interactions with TILs; (iv) types and sequences of NAC regimens; and (v) different proportions of estrogen receptor (ER)+ tumors in HER2+ breast cancer cohorts. In fact, the discrepancy in results regarding pre-NAC TIL levels and pCR in the immune response is attributable to the different NAC regimens based on anthracyclines and taxanes, including anthracycline or non-anthracycline regimens, HER2-targeted drugs with trastuzumab or trastuzumab plus pertuzumab, and lapatinib plus trastuzumab without chemotherapy (with the addition of endocrine therapy for ER+ tumors) [13,50,51]. In addition, immunohistochemical (IHC) staining has been used as a method to quantify TILs and immune cells [13,50], but multiplex immunofluorescence staining for immune cells and multispectral imaging characterizing immune infiltration facilitate the analysis of individual immune cell subpopulations and immune cell profiles associated with treatment response [51]. The lack of correlation between pre-NAC sTILs and pCR may be due to the relatively small sample size. However, an increase in sTIL in tumor biopsies during treatment was associated with non-pCR, whereas a decrease in the number of CD4+ and CD8+ T cells after treatment was associated with pCR [57]. The clinical significance of a transient increase in sTILs in patients who achieved pCR during treatment tumor biopsy is unclear. However, trastuzumab-based therapies may have different effects on the immune microenvironment within tumors [57]. The increase in TIL in RD after NAC is associated with progressive disease, and this may be due to the fact that immune cells were unable to recognize tumor cells in the immunosuppressive tumor microenvironment, thereby inhibiting the effective activation of antitumor immune responses [13].

Differences in the association between pre-NAC TIL levels and pCR have been reported between the TN and HER2+ subtypes [13], between the TN and luminal subtypes [14], and across all subtypes [12]. High pre-NAC TIL levels in HER2+ and luminal subtypes tended to be associated with higher pCR rates; however, these results were not statistically significant due to the sample size and small number of pCR cases in luminal type [13,14]. Regarding tumor responses to NAC, the importance of iTILs in the luminal subtype and the importance of both iTILs and sTILs in the TN subtype has been highlighted [9]. Other studies have suggested the importance of iTILs and sTILs in the TN and HER2+ subtypes but not in the luminal subtype [10]. Despite the relatively small number of cases showing pCR, it is unclear why only the luminal type shows the characteristic of significant iTIL/insignificant sTIL. This subtype may have unique biological factors that interact with tumor lymphocytes but not with stromal lymphocytes [84]. Thus, the exact role of iTILs and sTILs in the shrinkage of different breast cancers by NAC remains unclear. The conflicting results regarding the role of iTILs and sTILs in breast cancer may be attributed to the following factors: (i) heterogeneity in study designs, (ii) diversity in the methods and criteria used to quantify TILs, and (iii) interobserver variability in TIL assessment.

Higher pre-NAC CD4+ and CD8+ T cell levels are associated with better survival rates, such as DFS and OS, in all subtypes [30,31,67]. However, contradictory results have been reported for post-NAC changes in TIL levels. In the TN subtype, a decrease in TIL levels has been shown to be associated with improved DFS [49]. A decrease in TIL levels has been shown to be associated with local recurrence [30]. In contrast, an increase has been shown to be associated with worse RFS and OS [38], but it has also been shown to be associated with better distant DFS in RD [48]. The reason why the increase in TIL after NAC had a negative impact on survival may be because different NAC regimens have different effects on the immune response. Persistent high TIL levels after NAC may indicate the onset of an unbalanced immune response in which immunosuppressive infiltration predominates, reflecting an overall profile of chemotherapy resistance [38]. On the other hand, the increase in TILs is thought to be due to NAC inducing the release of DAMPs through the killing or damage of cancer cells. DAMPs activate immune cells, alter the tumor microenvironment, promote the generation of tumor-specific novel antigens within cancer cells, and recruit immune cells [49]. NAC induces increased T cell infiltration, T cell repertoire clonality, and increased inflammatory signatures, thereby converting cold tumors into hot tumors and promoting a more inflammatory tumor immune microenvironment [48].

In the HER2+ type, an increase in post-NAC TIL levels has been shown to be associated with improvement in BCSS, DFS [54], and EFS [64], whereas a decrease in TIL levels may indicate local recurrence [30]. NAC with dual HER2 inhibition is widely used as a chemotherapy regimen. This therapy improves immune evasion by tumor cells, increases sensitivity to CTLs, induces immunogenic cell death in tumor cells, and suppresses Tregs and MDSCs, thereby improving host immune evasion [30]. A decrease in TILs may be associated with recurrence and play a role in tumor recurrence by reducing the immune response to tumor cells in the tumor microenvironment. However, conflicting results have been reported, suggesting that high NAC TIL levels in RD after NAC in HER2 type are associated with poor prognosis because the microenvironment of the RD exhibits immunosuppressive characteristics [15]. Incomplete eradication of cancer cells by chemotherapy and anti-HER2 targeted therapy, coupled with high levels of immune cell infiltration, reflects a self-perpetuating vicious cycle driven by harmful interactions between cancer cells and immune cells, potentially skewing toward an immunosuppressive/tumor-promoting polarization [15]. In the luminal subtype, higher levels of CD8+ TIL and low expression of FOXP3 have been shown to be associated with better DFS and OS [66,66]. Thus, CD8+ TILs play an important role in prolonging survival, but the functional significance of changes in survival due to decreases or increases in TIL levels after NAC may differ depending on the tumor subtype. In the luminal subtype, an increase in TILs after NAC are a poor prognostic factor for survival, suggesting that immune cell infiltration in this subtype has different biological characteristics, such as M2 macrophages [12].

Overall, the results regarding predictive and prognostic factors for TIL in breast cancer subtypes are summarized in Figure 5. High pre-NAC TIL levels led to pCR, which was associated with favorable prognosis across all subtypes. Conversely, high post-NAC TIL levels were associated with either favorable or poor prognosis in TN and HER2 types, but in the luminal type, high post-NAC TIL levels were associated with poor prognosis. In contrast, low post-NAC TIL levels were associated with favorable prognosis in the luminal type, but low post-NAC TIL levels were associated with poor prognosis in the TN and HER2 types.

### 6.2. Predictive and Prognostic Values of PD-L1, FOXP3, and Systemic Immune Response

Tumors that respond to NAC have been shown to have higher TCR clonality in CD8+ T cells compared to tumors that are resistant to NAC, suggesting that CD8+ T cells play an important role in regulating the local antitumor immune state and enhancing NAC response [37]. High TCR clonality is associated with PD-L1 positivity, which is activated through increased IFN-γ production from activated T cells. In PD-L1-positive TN breast cancer, improved sensitivity to NAC and high pCR rates were observed [37]. Regarding combination therapy with PD-L1 inhibitors, NAC improves pCR rates and survival compared to chemotherapy alone in unselected TN breast cancer [37]. HER-2 is an important target for trastuzumab, inducing PD-L1 expression and promoting T cell recruitment and activation. This suggests that the efficacy of trastuzumab is associated with TILs and PD-L1 expression [58]. Trastuzumab induces antibody-dependent cell-mediated cytotoxicity via immune cells, promotes immunogenic cell death, and enhances antitumor immune responses through the increase in CD8+ T cells [58].

FOXP3-positive T cells (classified as Tregs) exert an immunosuppressive effect against tumors in the tumor microenvironment. These cells constantly express PD-1 and cytotoxic T lymphocyte-associated protein 4 (CTLA-4), thereby suppressing the activity of CTLs and DCs. Tregs produce cytotoxic substances such as perforin and granzyme, thereby inhibiting CTL function [40]. CTLs and Tregs have opposing effects on tumor immunity in the tumor microenvironment. Therefore, the CD8/FOXP3 ratio (CFR) was a reliable biomarker for predicting response to NAC and prognosis in TN breast cancer, but this trend was not observed in luminal breast cancer [40]. CFR decreased in recurrent TN breast cancer and HER2+ breast cancer but did not change in the luminal type [41]. The decrease in CFR may be involved in these recurrences, and the decline in tumor immunity in the tumor microenvironment plays a role in recurrence. On the other hand, tumors with low Treg infiltration are considered to have a high probability of achieving pCR, while tumors with high Treg infiltration have been associated with pCR [73]. Some chemotherapy agents, particularly cyclophosphamide, have been reported to suppress Tregs. The suppression of Treg function by chemotherapy is more pronounced in tumors with high Treg infiltration levels, which may promote tumor attack by CTLs and contribute to the achievement of pCR [73].

NLR values showed a significant negative correlation with TIL density. In other words, the higher the TIL density, the lower the NLR, suggesting that there is interaction between immune cells and that NLR may reflect the infiltration of immune cells into the tumor stroma [28]. TILs in patients with high NLR are mainly composed of Tregs. Neutrophils promote Treg differentiation and induce apoptosis of CD8+ T cells, so local immune responses may be suppressed in patients with high NLR [28]. Multiple factors, including NLR, CD8+ TILs, and Tregs, are associated with local and systemic immune responses to NAC in breast cancer. A significant correlation was observed between low NLR and high pCR rates, suggesting that NLR is an independent predictive factor of pCR [28]. High NLR prior to treatment was independently associated with worsened OS in TN and HER2+ breast cancer, but no association was observed in luminal breast cancer [28]. Patients with low NLR and high CD8+ TIL infiltration showed improved survival rates; however, only NLR was identified as an independent prognostic factor for TN, and patients with low NLR had longer survival periods [28]. Similarly, high TIL levels, low NLR, and low PLR were associated with pCR. Low NLR independently predicted pCR, and TIL, NLR, and PLR predicted the efficacy of NAC in TN breast cancer [79]. Similarly to other systemic immune responses associated with NAC, those with TN breast cancer who achieved pCR had significantly higher plasma concentrations of proinflammatory cytokines, growth factors, and Th2 cytokines compared to patients who did not achieve pCR. Systemic cytokines and growth factors measured at diagnosis may function as independent biomarkers for pCR. A strong association was observed between systemic immune mediators and pCR, TIL intensity, and PFS [39].

### 6.3. Evaluation of TILs

When performing overall visual assessments of TILs, pathologists’ ability to distinguish the exact distributions of iTILs and sTILs in tumors and to define appropriate cut-off percentages may be limited due to tumor heterogeneity. In addition, CD4+, CD8+, and FOXP3+ T cells and macrophages cannot be distinguished by hematoxylin and eosin (H&E) staining. As H&E staining reveals only the extent of total lymphocyte infiltration, preventing immune cell-level determination of biological and functional roles, and as TIL populations contain antitumor and protumor immune cells, standardized sTIL or iTIL assessment according to the International TILs Working Group guidelines is recommended [8,85].

The classification of T cells, NK cells, B cells, and macrophages by antibodies to CD3, CD4, CD8, FOXP3, CD56, CD20, CD68, and CD163 has been based on single markers in conventional IHC analysis, but the visual evaluation of TILs is biased and not reproducible among pathologists [86,87]. Single-plex IHC staining of serial sections enables the quantification of infiltration by specific immune cell populations but is insufficient to characterize their functional statuses in breast cancer specimens. In contrast, multiplex techniques (e.g., IHC and immunofluorescence) enable the simultaneous phenotyping of multiple immune cell populations [88]. CD3+ CD8+ T cells representing CTLs and CD3+ CD4+ FOXP3+ cells representing Tregs with different biological functions in the tumor microenvironment can be characterized simultaneously. In addition, the evaluation of immune cells in a multiplexed environment can enhance the specific mapping of immune cells in relation to tumor cells. Furthermore, when combined with digital image analysis, the computational assessment of TILs allows for quantification of TILs in the stroma within and around tumors more rapidly and with higher throughput than manual assessment. The standardized computational assessment of TILs would solve many of the problems plaguing visual TIL assessment and would be a promising, more reliable method to complement local TIL evaluation [88]. In addition, AI (artificial intelligence)-based classifiers can incorporate pathologists’ domain-specific knowledge in the development of image analysis algorithms. AI classification methods are now used widely for tissue segmentation and lymphocyte detection on images of H&E-stained specimens. AI-based approaches have shown better or comparable accuracy relative to pathologists’ assessments. The integration of AI algorithms into multiplex image analysis improves immune cell identification and phenotypic separation [88]. Good agreement has been observed between visually and automatically obtained TIL scores. Further testing and validation in this area are needed.

### 6.4. Comprehensive Understanding of Antitumor Immune Responses

As alternatives to conventional IHC, several comprehensive analyses of TIL roles in therapeutic responses have been performed; they have involved the estimation of immune cell activity [61]; examination of multiple B cell gene signatures [60], T cells, and immune cell aggregation in HER2+ breast cancer [62]; metagenetic analysis of B, plasma, T, NK, and myeloid/dendritic cells [75] and stromal AQUA for CD3, CD8, and CD20 cells in NS breast cancer [83]; and examination of immune-related gene signatures and gene expression in luminal breast cancer [65]. Other proposed markers of tumor response and survival include increased PD-L1 expression in TN breast cancer [37] and pre-NAC PD-1 and PD-L1 expression in NS breast cancer [26,74]. For immunoprofiling studies performed with large samples, tissue microarrays, instead of whole sections, can be used; in some cases, however, the microarray tissue cores do not adequately represent the heterogeneity of marker expression in single samples.

Reductions in the NLR and PLR, which reflect the systemic immune response, have been found to be important for therapeutic effects in TN [28] and NS [79] breast cancers and for TN breast cancer survival [28]. The increase in lymphocytes relative to neutrophils and monocytes in the peripheral blood evokes a systemic immune response, which may work in concert with antitumor immune activation in the tumor microenvironment to enhance therapeutic efficacy and prolong survival in patients with breast cancer. The upregulation of antitumor factors by CD4+ and CD8+ T cells and the downregulation of protumor factors by Tregs, as local immune responses in the tumor microenvironment, as well as increased peripheral NK cell activity and decreased PLRs as systemic immune responses, have been found to be necessary for post-NAC therapeutic effects in patients with breast cancer [89,90]. These findings suggest that the coactivation of local and systemic immune responses is important for the therapeutic effect of NAC.

### 6.5. Unresolved Issues and Strategies to Enhance Antitumor Immune Responses for Therapeutic Effects and Survival

Persisting issues in the examination of antitumor immune responses in patients with breast cancer are the mechanism of antitumor immunity activation and its contribution to therapeutic effects and survival. CD8+ T cells play important roles in evoking antitumor immunity, in terms of the local power balance of protumor factors in the tumor microenvironment and of systemic coactivation with increased peripheral lymphocyte levels and NK cell activity. The activation of the antitumor immune response and downregulation of the protumor immune response, as with anticancer drugs, enhance therapeutic effects, and the production of memory T cells recognized by neoantigens such as TAs and TAAs can eradicate residual cancer cells, prolong survival, and, ultimately, cure cancer. Immune checkpoint inhibitors (ICIs), which represent a major recent advance in cancer immunotherapy [91,92], inhibit the downregulation of PD-1-, CTLA-4-, and PD-L1-mediated immune activation in immune and tumor cells. The introduction of ICIs such as anti-PD-1 and anti-PD-L1 antibodies has increased breast cancer treatment effects, especially when these agents are used in combination with anticancer drugs for TN breast cancer [93]. Research on ICIs indicates that the release of the immunosuppressive state reconstituted by TDSFs for tumor progression is more important for the enhancement of the tumor response than it is for the conventional stimulation of an antitumor immune response. Although ICIs act on the tumor microenvironment in a tumor-specific manner, their use has critical drawbacks, as their effects interfere with the regulation of autoimmune reactions and cause severe systemic side effects [immune-related adverse events (irAEs)] [93]. The combined treatment of breast cancer with ICIs and anticancer drugs increases the pCR rate, but whether this increase is associated with better survival remains unclear.

Another potential strategy to block immunosuppression in the tumor microenvironment is targeted therapy against MDSCs, Tregs, and M2 macrophages. MDSCs contain immature myeloid cells, including immature DCs, neutrophils, and monocytes, and the targeting of these individual cell types may be difficult. Anti-CTLA-4 antibodies are used to target Tregs, but they are not expected to improve therapeutic effects when used in combination with anticancer agents due to irAE occurrence [94]. M2 macrophages express signal regulatory protein α (SIRPα), which interacts with CD47, the “don’t eat me” signal expressed on tumor cells [95]. Anti-CD47 antibodies inhibit CD47–SIRPα interaction, allowing antibody-dependent cell phagocytosis after macrophage checkpoint inhibition [96]. The efficacy of combination therapy with anti-CD47 and anti-CD20 antibodies for B cell and follicular lymphomas was demonstrated in a phase Ib trial [96]. M2 macrophage infiltration has been reported to adversely affect breast cancer prognosis [97]. Combination therapy with anti-CD47 antibodies and trastuzumab may enhance antitumor immunity, as evidence suggests that antibody-dependent cytotoxicity plays an important role in the activity of trastuzumab. As M2 macrophages play important roles in breast cancer progression and metastasis [98], anti-CD47 antibodies may have promising therapeutic effects on breast cancer. Furthermore, although the genetic instability of tumor cells is a key factor determining the response to T cell–targeted checkpoint inhibitors, the effects of macrophage checkpoint inhibitor therapy are independent of neoantigen recognition. Thus, strategies involving combined T cell and macrophage checkpoint inhibition may have synergistic effects. Such effects have been demonstrated with the use of anti-CD47 antibodies with chemotherapy and ICIs in preclinical models of breast cancer [99]. Additional clinical trials are needed to determine the clinical efficacy of these strategies while safely resolving issues of toxicity, such as anemia, thrombocytopenia, and leukopenia.

The role of NAC varies according to the breast tumor subtype and is less clear for luminal breast cancer than for TN and HER2+ breast cancers [100]. In clinical practice, several exceptions have been observed; for example, TN breast cancer does not respond to anthracycline- and taxane-based regimens even in the presence of TILs, whereas luminal breast cancer responds to NAC regardless of the TIL level. The factors underlying the difference in response to NAC between the TN and luminal subtypes are unknown. The presence of TILs has more prognostic than predictive value for the response to chemotherapy. The determination of the TIL immunophenotype is important for the understanding of these cells’ roles, but the key factors for pCR achievement have not been definitively identified. The coactivation of local and systemic immune responses is known to contribute to post-NAC tumor responses, and activated antitumor immunity is known to prolong the survival of patients with breast cancer. The host immune response to tumor cells may be an individual-level limiting factor, whereas the drug sensitivity of tumor cells is a determining factor in the enhancement of antitumor immunity. Increased tumor cell death, such as via ICD and pyroptosis, activates an antitumor immune response that is responsible for tumor shrinkage and survival.

Differences in antitumor immunity among breast cancer subtypes have traditionally been characterized by the statement that breast cancer is generally less immunogenic, but TN breast cancer is more immunogenic than the luminal type. ER-negative tumors have a higher mutational load than ER-positive tumors and may express high levels of neoantigens that cause a strong immune response [101]. Another estrogen-related form of immunomodulation that affects antitumor immunity is immunosuppression via the stimulation of the release of the tumor-promoting cytokine CCL5 in the tumor microenvironment, as in basal type breast cancer [102,103]. In addition, the immunosuppressive state mediated by the immune phenotypes of TILs such as M2 macrophages and Tregs in ER-positive tumors may reduce tumor responsiveness and worsen prognosis more than in the TN and HER2+ subtypes [104]. However, the achievement of pCR after NAC is more likely for luminal A tumors, characterized by a low grade, little proliferation, strong ER expression, and a small percentage of TILs, than for luminal B tumors with a high grade, abundant proliferation, weak ER expression, and a large percentage of TILs [105]. The large number of TILs in luminal B tumors may be attributable to the strong expression of neoantigens required for the antitumor immune response due to these tumors’ high clonal diversity and mutational load [106]. This is associated with the biologically aggressive characteristics listed above, as well as increased lymph node metastasis and tumor size, but stronger post-NAC immune activation leading to greater therapeutic efficacy than for luminal A tumors can be expected [107]. The clinical relevance and significance of TIL levels in luminal breast cancer need to be evaluated further.

## 7. Conclusions

The conflicting, paradoxical, and inconsistent results reported for TIL-related predictive and prognostic factors can be explained by factors such as the use of different technical methods and cut-off values, as well as different NAC regimens and immune responses, and by tumor heterogeneity and sample size. Because TILs include mononuclear immune cells, the determination of pre- and post-NAC TIL immunophenotypes in patients with breast cancer is important. The functional roles of immune cells in the tumor microenvironment and peripheral blood contribute to tumor shrinkage and improved patient survival.

Accumulating evidence indicates that the substantial activation of TIL-mediated antitumor immune responses is required for post-NAC breast tumor shrinkage, regardless of the tumor subtype. The functional roles of TILs depend on the tumor microenvironment, but a strong immune response is not always necessary to achieve pCR. Tumor sensitivity to anticancer drugs, cell death induced by those drugs, and the subsequent activation of antitumor immunity determine the tumor response and prognosis. The achievement of pCR requires the activation of an effective antitumor immune response that eradicates residual tumor cells after NAC, leading to the curing of breast cancer. The mechanisms by which immune responses induced by anticancer drugs eliminate cancer cells and prolong survival need to be investigated further, with an improved understanding of the heterogeneity of breast tumor immunogenicity in relation to TIL functional subtypes and the modulation of the tumor microenvironment.

Multiple strategies exist for breast cancer patients undergoing NAC regarding the clinical significance of enhancing pCR and improving survival rates through pre-, intra-, and post-NAC TIL assessment. If changes in TIL are confirmed as a predictor of pCR, early assessment of TIL levels during chemotherapy could serve as an early alternative indicator of treatment resistance, suggesting the possibility of earlier switching to secondary treatment. In tumors failing to achieve pCR, lymphocyte infiltration after NAC was associated with high residual tumor burden and poor clinical outcomes. It is crucial whether the lymphocyte infiltration remaining in non-responsive tumors is enriched with immunosuppressive cells that could be targets for immune checkpoint inhibitors. Patients who fail to achieve pCR and have high TILs after NAC should be enrolled in specific secondary treatment trials.

Even though immune gene signatures and immune cell infiltration do not appear to be universally associated with tumor response or outcome in patients with breast cancer treated with NAC, the key factors of antitumor immunity that ultimately lead to the curing of breast cancer need to be elucidated. Further functional analyses of TIL immunophenotypes according to breast cancer subtypes and the establishment of a reproducible, objective, and accurate method for the evaluation of TILs in patients with breast cancer are needed.

## Figures and Tables

**Figure 1 ijms-26-09959-f001:**
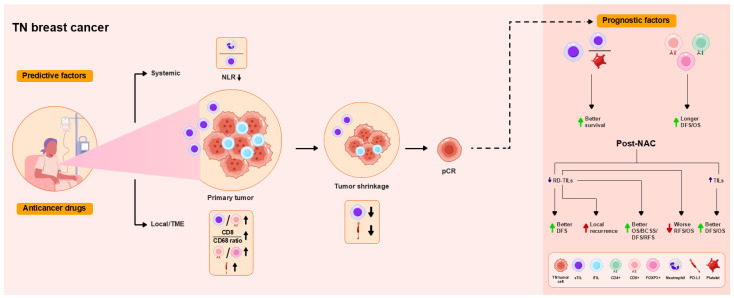
Overview of the functional roles of tumor-infiltrating lymphocytes (TILs) as predictive and prognostic factors in patients with triple-negative (TN) breast cancer treated with neoadjuvant chemotherapy (NAC). Increased levels of stromal tumor-infiltrating lymphocytes (sTILs), reflected by the pre-NAC cluster of differentiation (CD)8+ T cell level, programmed death ligand 1 (PD-L1) expression, and CD8/CD68 ratio, are associated with tumor shrinkage, leading to pathological complete response (pCR) locally in the tumor microenvironment (TME), consistent with decreases in the sTIL level and PD-L1 expression in tumor cells. High TIL levels and high CD8/FOXP3 ratios are associated with pCR. Intratumoral TIL (iTIL) and sTIL levels predict pCR. Low neutrophil-to-lymphocyte ratios (NLRs) are associated with pCR as a systemic immune response. pCR is associated with improved DFS and OS. High TIL levels and low platelet-to-lymphocyte ratios (PLRs) before NAC predict better survival. Higher pre-NAC CD4, CD8, and forkhead box P3 (FOXP3) levels are associated with longer disease-free survival (DFS) and overall survival (OS). Decreased TIL levels in residual disease (RD) after NAC are associated with improved DFS, as well as local recurrence. Increased TIL levels in RD are associated with improved OS, breast cancer-specific survival (BCSS), distant DFS, and recurrence-free survival (RFS), in addition to worse RFS and OS. Higher post-NAC TIL levels are associated with improved DFS and OS.

**Figure 2 ijms-26-09959-f002:**
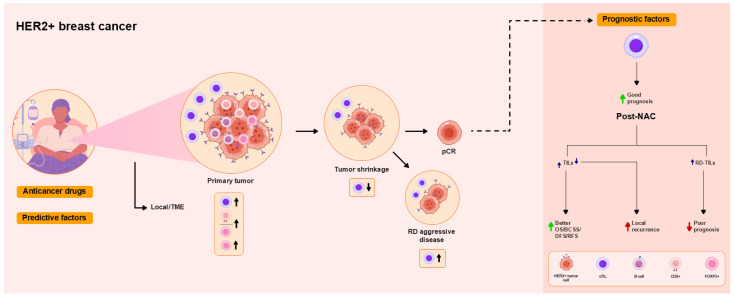
Overview of the functional roles of TILs as predictive and prognostic factors in patients with human epidermal growth factor receptor 2 (HER2)+ breast cancer treated with NAC. High TIL levels are associated with pCR. High CD8/FOXP3 ratios are associated with pCR. High pre-NAC TIL levels, reflecting the local immune response in the TME, tend to be or are not associated with pCR. Neither iTIL nor sTIL levels are predictors of pCR. Increased pre-NAC levels of sTILs [CD8+ T cells, B cells, and regulatory T cells (Tregs)] are associated with pCR, as was a decrease in sTILs, while increased post-NAC sTIL levels in RD are associated with aggressive disease. pCR is associated with improved DFS and OS. Higher pre-NAC TIL levels are associated with better prognosis, including improved BCSS, DFS, and event-free survival (EFS). Higher post-NAC TIL levels are associated with improved DFS and OS. Decreased post-NAC TIL levels are associated with local recurrence. Increased TIL levels in RD are associated with worse prognosis.

**Figure 3 ijms-26-09959-f003:**
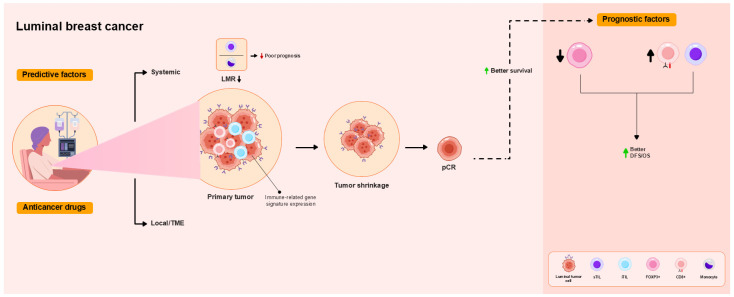
Overview of the functional roles of TILs as predictive and prognostic factors in patients with luminal breast cancer treated with NAC. High TIL levels are associated or tend to be associated with pCR. iTIL levels predict pCR. Pre-NAC intratumoral (i)CD8+ T cells and immune-related gene signature expression are associated with pCR as a local immune response in the TME. pCR is not associated with improved DFS and OS. Low FOXP3 and high iCD8 TIL levels are associated with better DFS and OS. Lower lymphocyte-to-monocyte ratios (LMRs) are associated with poor prognosis.

**Figure 4 ijms-26-09959-f004:**
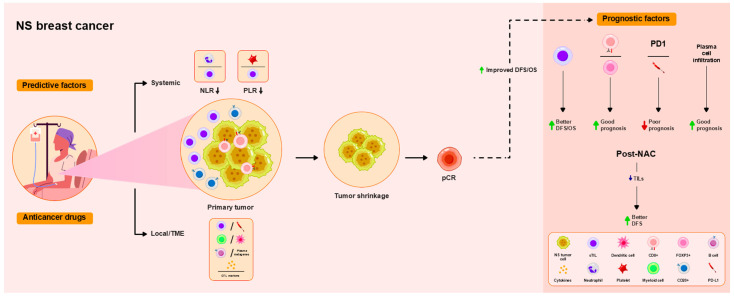
Overview of the functional roles of TILs as predictive and prognostic factors in patients with non-subtype-specified (NS) breast cancer treated with NAC. As a local immune response in the TME, high TIL levels are associated with pCR in all subtypes. Levels of stromal B (sCD20) and CD8+ T cells, chemokines, and cytotoxic T lymphocyte (CTL) markers are associated with pCR. Myeloid/dendritic cells and B cells/plasma metagenes are associated with good pathological response. Lower NLRs and PLRs, representing the systemic immune response, are associated with pCR. Higher pre-NAC TIL levels are associated with improved DFS, and lower post-NAC TIL levels are associated with improved DFS. High CD8/FOXP3 ratios predict good prognosis, whereas PD-L1 and PD-1 expression correlate with poor prognosis. Plasma cell infiltration correlates with the prolonged DFS of patients with HR-negative breast cancer.

**Figure 5 ijms-26-09959-f005:**
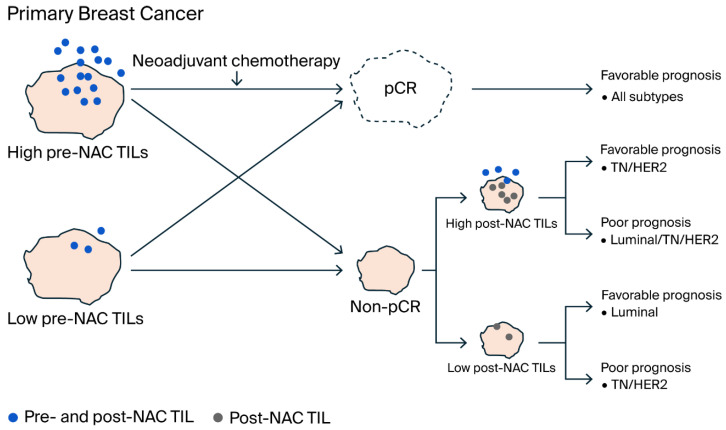
Schematic representation of the mechanism by which TILs correlate with pCR and prognosis in breast cancer subtypes. Patients with high pre-NAC TIL levels tended to achieve pCR across all subtypes and had favorable prognosis. In contrast, non-pCR patients who did not achieve pCR but had high post-NAC TIL levels showed either favorable or poor prognosis in TN and HER2-types. In the luminal type, high post-NAC TIL levels indicated poor prognosis, while low post-NAC TIL levels indicated favorable prognosis. In the TN and HER2 types, low post-NAC TIL levels indicated poor prognosis. The conflicting prognosis associated with high TIL levels after NAC in the TN and HER2 subtypes may be due to RD microenvironment imbalances affecting TIL-related immunosuppressive or immune active characteristics. Even with low TIL levels prior to NAC, effective immune activation through chemotherapy or targeted therapy may lead to pCR.

**Table 1 ijms-26-09959-t001:** Clinical significance of TILs and antitumor immune responses for pCR and RD after NAC in TN breast cancer.

Ref. (Year)	Subtype	TILs/Other	Immunophenotype/ Gene and Protein Expression	Predictive Factors	Prognostic Factors
[27] (2024)	TN	TILs	ND	TILs associated with pCR after NAC+ICI	ND
[47] (2023)	TN	RD TILs	CD4, CD8	ND	Increased OS/BCSS with high TIL levels in RD after NAC
[28] (2023)	TN	TILs/NLR	CD8, FOXP3	Low NLR/high TIL density associated with high pCR rate	Low NLR/high TIL levels associated with improved OS/BCSS
[39] (2023)	TN	sTILs/immune mediators	ND	Systemic immune mediators correlated with high TIL levels, leading to pCR	Circulating cytokines with TILs predict improved survival
[29] (2023)	TN	TILs	PD-L1	High TIL levels associated with pCR; PD-L1+ associated with higher pCR rate after anticancer drug+ICI	ND
[40] (2023)	TN	TILs	CD8/FOXP3 ratio, PD-1, PD-L1	CD8/FOXP3 ratio only independent predictor of pCR	ND
[48] (2023)	TN	sTILs	ND	Higher pCR rate with higher TIL levels, Carb incorporation into A-T	Increased TIL levels in RD associated with distant DFS for A-T, A-T/Carb
[38] (2022)	TN	TILs	ND	Change in TIL levels with treatment correlated with metabolic response (SUV), pCR	High post-NAC TIL levels predicted poor RFS/OS
[35] (2022)	TN	TILs	ND	Patients aged <40 years had higher TIL levels, pCR rate	ND
[45] (2021)	TN	TILs/PLR	ND	TIL levels retained pCR predictive value	PLR better predictor of less distant recurrence or longer distant RFS with pCR or small RD
[37] (2021)	TN	TILs	TCRs, PD-L1, CD3/CD68 ratio	Higher TCR clonality, PD-L1+, higher CD3/CD68 ratios, closer T/tumor cell proximity associated with pCR; T cell proximity, PD-L1 expression enhanced pCR prediction	ND
[25] (2021)	TN	TILs	PD-L1, TIM-3, LAG-3	TIL levels, PD-L1 expression decreased after NAC in pCR; TIM-3+ more frequent in non-pCR	High TIL levels associated with better OS; PD-L1 expression, TIM-3+ associated with worse OS
[46] (2020)	TNI	TILs	ND	ND	TIL levels associated negatively with EFS
[42] (2020)	TN	sTILs	ND	sTIL levels associated with RCB 0/I status, not pCR; HRD associated with pCR, RCB 0/I status	ND
[49] (2020)	TN	TILs	ND	ND	Decrease in TIL levels improved DFS
[32] (2019)	TN	TILs	ND	High TIL levels associated with diffuse TIL pattern, high pCR rate	Higher TIL density, diffuse TIL pattern correlated with improved RFS/OS
[43] (2019)	TN	TILs	ND	RD TIL levels correlated with CD8 T cell density	Higher RD TIL levels associated with improved RFS/OS, especially in RCB II
[33] (2018)	TN/HER2+	TILs	ND	TILs as strong predictors of pCR	ND
[30] (2018)	TN/HER2+	TILs	ND	High TIL levels associated with pCR	High TIL levels predict good prognosis; decreased TIL levels potentially associated with local recurrence
[36] (2017)	TN	TILs	ND	pCR rate higher for LPBC than non-LPBC	DFS better for LPBC than non-LPBC
[31] (2016)	TN	TILs	CD3, CD4, CD8, FOXP3	Higher pre-NAC TIL levels, CD8/CD4 ratio, post-NAC CD4 level associated with pCR	Longer DFS associated with higher pre-NAC CD3, CD4, CD8, FOXP3 levels, CD4/FOXP3 ratio; longer OS associated with higher pre-NAC CD3 level
[34] (2012)	TN	TILs/apoptosis	ND	Higher TIL levels, apoptosis scores associated with pCR	ND

TIL—tumor-infiltrating lymphocyte; TN—triple-negative; pCR—pathological complete response; NAC—neoadjuvant chemotherapy; Ref.—reference; ND—no data; ICI—immune checkpoint inhibitor; RD—residual disease; CD—cluster of differentiation; OS—overall survival; BCSS—breast cancer-specific survival; NLR—neutrophil-to-lymphocyte ratio; FOXP3—forkhead box P3; sTIL—stromal tumor-infiltrating lymphocyte; PD-L1—programmed cell death ligand 1; PD-1—programmed cell death 1; Carb—carboplatin; A-T—anthracycline and taxanes; DFS—disease-free survival; SUV—standardized uptake value; RFS—relapse-free survival; PLR—platelet-to-lymphocyte ratio; TCR—T cell receptor; TIM-3—T cell immunoglobulin mucin-3; LAG-3—lymphocyte activation gene 3; TNI—triple-negative inflammatory; EFS—event-free survival; RCB—residual cancer burden; HRD—homologous recombination deficiency; HER2—human epidermal growth factor receptor 2; LPBC—lymphocyte-predominant breast cancer.

**Table 2 ijms-26-09959-t002:** Clinical significance of TILs for pCR and RD after NAC in HER2+ breast cancer.

Ref. (Year)	Subtype	TILs/Other	Immunophenotype/ Gene and Protein Expression	Predictive Factors	Prognostic Factors
[50] (2024)	HER2+	TILs	ND	TIL levels not associated with pCR; trend of association for HR+	Excellent 3-year invasive DFS, regardless of pCR
[52] (2024)	HER2+	TILs	ND	TIL levels associated with pCR, numerical or high levels predict pCR	ND
[53] (2024)	HER2+	TILs	CD8	High TIL levels, CD8 infiltration predict pCR	ND
[15] (2023)	HER2+	RD TILs	ND	ND	Higher RD TIL levels associated with poor prognosis
[61] (2023)	HER2−	TILs	iICA of 23 immune cell types	TIL levels correlated with iICA cluster, better predicted pCR	iICA cluster predicted better DFS/OS
[60] (2023)	HER2+	TILs/immune-related signatures	36 immune signatures	Multiple B cell-related signatures more associated with pCR than TIL levels	Multiple B cell-related signatures more associated with EFS than TIL levels
[62] (2022)	HR+/HER2+	sTILs	CD3+, CD3+CD8−FOXP3−, CD8+, FOXP3+	sTILs were associated with pCR; increased CD3+, CD3+CD8−FOXP3−, CD8+, FOXP3+ sTIL levels, immune cell aggregates associated with pCR in TME	ND
[58] (2022)	HER2+	TILs	PD-L1	High TIL, PD-L1+ TIL levels predicted pCR; PD-L1+ tumor cells reduced by Tz + chemotherapy	ND
[57] (2021)	HER2+	TILs	TILs, sTILs, CD4	pCR associated with higher TIL (not sTIL) levels pre-treatment, higher sTIL (not TIL) levels during treatment; infiltrative lymphocyte levels increased in non-pCR, CD8 levels decreased in pCR cases during treatment	ND
[59] (2020)	HER2+	TILs, Ki-67	CD8, FOXP3	High Ki-67 levels strongly predict pCR; TIL, and FOXP3+ T cell levels may play roles in tumor response	ND
[51] (2020)	HER2+	sTILs	CD4, CD8, CD20, CD68, FOXP3	High CD4+, CD8+, CD20+ sTIL, CD20+ iTIL levels associated with higher pCR rates; pCR associated with more baseline sCD4, iCD4, iCD20+ TIL infiltration	ND
[54] (2019)	HER2+	TILs	ND	High pre-NAC TIL levels predict pCR	Increased post-NAC TIL levels associated with improved BCSS/DFS
[64] (2019)	HER2+	TILs	ND	Baseline TIL levels not associated with pCR	Increased baseline TIL levels associated with improved EFS
[33] (2018)	TN/HER2+	TILs	ND	TIL levels strongly predict pCR	ND
[30] (2018)	TN/HER2+	TILs	ND	High TIL levels associated with pCR	High TIL levels predicted good prognosis; decreased TIL levels potentially associated with local recurrence
[63] (2017)	HER2+	sTILs	ND	Magnitude of TIL level decrease associated with pCR; High post-NAC TIL levels in RD associated with aggressive disease	High post-NAC TIL levels in RD associated with worse outcomes
[55] (2017)	HER2+	TILs	ND	High baseline TIL levels associated with increased pCR	ND
[56] (2016)	HER2+	TILs	ND	LPBC increased, predicted pCR	TILs more prognostically relevant than pCR in triple-positive breast cancer

TIL—tumor-infiltrating lymphocyte; pCR—pathological complete response; RD—residual disease; NAC—neoadjuvant chemotherapy; HER2—human epidermal growth factor receptor 2; Ref.—reference; ND—no data; DFS—disease-free survival; CD—cluster of differentiation; iICA—inferred immune cell activity; OS—overall survival; EFS—event-free survival; sTIL—stromal tumor-infiltrating lymphocyte; FOXP3—forkhead box P3; HR—hormone receptor; TME—tumor microenvironment; PD-L1—programmed cell death ligand 1; Tz—trastuzumab; iTIL—intratumoral tumor-infiltrating lymphocyte; BCSS—breast cancer-specific survival; TN—triple-negative; LPBC—lymphocyte-predominant breast cancer.

**Table 3 ijms-26-09959-t003:** Clinical significance of TILs for pCR and RD after NAC in luminal breast cancer.

Ref. (Year)	Subtype	TILs/Other	Immunophenotype/ Gene and Protein Expression	Predictive Factors	Prognostic Factors
[65] (2022)	L-B	TILs/immune-related gene signatures, immune cell subpopulations	ND	TIL levels, immune-related gene signatures, immune cell subpopulations associated with pCR; sequential anthracyclines/anti-PD-1 may activate antitumor immune response in basal molecular subtype	ND
[68] (2020)	ER+/HER2−	TILs/LMR	ND	ND	TILs > 10%, LMR < 5.2 correlated with poor prognosis
[66] (2018)	HR+/HER2−	TILs/GE	FOXP3 T cells, CD163 macrophages	GE associated with higher pCR rates, tumor shrinkage. GE, TIL levels predicted tumor shrinkage	Low FOXP3 levels associated with improved DFS
[67] (2017)	L-B/HER2−	TILs	CD8	iCD8+ TIL level correlated with pCR	CD8+ TIL level correlated with DFS/OS; strong iCD8+ TIL expression associated with OS

TIL—tumor-infiltrating lymphocyte; pCR—pathological complete response; RD—residual disease; NAC—neoadjuvant chemotherapy; Ref.—reference; L-B—luminal B; ND—no data; PD-1—programmed cell death 1; ER—estrogen receptor; HER2—human epidermal growth factor receptor 2; LMR—lymphocyte-to-monocyte ratio; HR—hormone receptor; GE—gene expression; FOXP3—forkhead box P3; CD—cluster of differentiation; DFS—disease-free survival; OS—overall survival.

**Table 4 ijms-26-09959-t004:** Clinical significance of TILs for pCR and RD after NAC in NS breast cancer.

Ref. (Year)	Subtype	TILs/Other	Immunophenotype/ Gene and Protein Expression	Predictive Factors	Prognostic Factors
[69] (2024)	NS	TILs	Titin	Titin expression elevated in pCR	ND
[76] (2024)	NS	sTILs/iTILs	ND	Pre-NAC sTIL levels predict pCR regardless of tumor subtype	ND
[77] (2024)	NS	TILs	ND	High TIL levels associated independently with pCR	ND
[79] (2023)	NS	TILs/NLR, PLR	ND	TIL levels, NLR, PLR predict pCR	ND
[78] (2023)	NS	TILs	ND	Higher pre-NAC TIL levels predict pCR	Lower post-NAC TIL levels and pCR associated with DFS
[44] (2023)	NS	sTILs	ND	High TIL levels predict pCR regardless of subtype	pCR associated with improved DFS/OS in TN, HER2+ (not luminal) subtypes
[82] (2022)	NS	TILs	eTregs, CD4^+^FOXP3^high^CD45RA^−^, other FOXP3^+^ Treg subsets, CD8	CD8/eTreg ratio independently predicts pCR	ND
[5] (2021)	NS	TILs	ND	High pre-NAC TIL levels predict higher pCR rates	Pre-NAC TIL levels associated with DFS
[81] (2021)	NS	TILs/LNR	ND	TIL levels may predict pCR rate, postoperative residual lymph node involvement	TIL levels may predict DFS
[72] (2021)	NS	PCs, CD8, CD4, CD8 FOXP3, B cells	ND	PC and B cell infiltration correlated with pCR	PC infiltration correlated with longer DFS in HR− patients
[11] (2020)	NS	TILs	ND	High TIL levels predict pCR in TN, HER2-enriched (not luminal) subtypes	High TIL levels associated with better DFS/OS for TN, HER2-enriched subtypes, worse survival for luminal subtype
[13] (2019)	NS	sTILs	ND	Pre-NAC TIL levels higher in pCR cases for luminal, TN (not HER2+) subtypes; mean TIL levels decreased after NAC in association with pCR	Pre-NAC TIL levels associated with DFS; High post-NAC TIL levels associated with impaired DFS in HER2 (not luminal, TN) subtype
[71] (2019)	NS	TILs/CCL21, CCL19	CTL	Higher baseline TIL, chemoattractant cytokine (CCL21, CCL19), CTL marker levels associated with higher pCR rates; stromal functions associated with RD; TIL levels, most immune gene expression decreased after NAC in pCR cases	ND
[26] (2018)	NS	TILs/PD-L1	ND	Higher baseline TIL levels, PD-L1+ rate associated with higher pCR rate; TIL levels (not PD-L1 expression) decreased after NAC	ND
[12] (2018)	NS	TILs	ND	Increased TIL levels predicted response to NAC in all subtypes	Increased TIL levels associated with improved DFS in TN, HER2+ (not luminal) subtypes, improved OS in TN (not HER2+) subtype, shorter OS in luminal subtype
[14] (2018)	NS	TILs	ND	Higher TIL levels associated with higher pCR rates in TN subtype, tended to be associated with pCR in luminal subtype	Low post-NAC TIL levels associated with better RFS for luminal subtype
[9] (2018)	NS	sTILs/iTILs	ND	pCR predicted by iTIL (not sTIL) levels in luminal subtype, iTIL and sTIL levels in TN subtype, neither in HER2+ subtype	ND
[74] (2017)	NS	TILs/PD-L1, PD-1	ND	PD-L1, PD-1 expression correlated with increased TIL levels, pCR	PD-L1, PD-1 expression correlated with poor prognosis
[70] (2016)	NS	TILs	CD4, CD8, FOXP3	High TIL levels associated with improved pCR rates, especially in TN subtype; CD8 (not CD4, FOXP3) infiltration associated with higher pCR rates	High TIL levels associated with longer DFS/OS
[10] (2016)	NS	iTILs/sTILs	ND	iTIL, sTIL levels correlated with pCR in HER2+, TN subtypes; no biomarker correlated with pCR in luminal subtype	ND
[41] (2016)	NS	TILs	CFR	High CFRs associated with higher pCR rates in TN and HER2+ subtypes	High CFRs predict good prognosis
[80] (2015)	NS	iTILs/sTILs/PD-L1	ND	iTIL levels, sTIL levels, strong PD-L1 expression predict pCR	ND
[75] (2014)	NS	ND	B/P, T/NK, M/D metagenes	B/P, M/D, to lesser extent T/NK metagenes associated with favorable pathological responses	ND
[83] (2014)	NS	sTILs	CD3, CD8, CD20	TIL levels, stromal AQUA scores for CD3, CD8, CD20 predicted pCR	ND

TIL—tumor-infiltrating lymphocyte; pCR—pathological complete response; RD—residual disease; NAC—neoadjuvant chemotherapy; NS—non-subtype-specified; Ref.—reference; ND—no data; sTIL—stromal tumor-infiltrating lymphocyte; iTIL—intratumoral tumor-infiltrating lymphocyte; NLR—neutrophil-to-lymphocyte ratio; PLR—platelet-to-lymphocyte ratio; DFS—disease-free survival; OS—overall survival; TN—triple-negative; HER2—human epidermal growth factor receptor 2; eTreg—effector regulatory T cell; CD—cluster of differentiation; FOXP3—forkhead box P3; LNR—lymphocyte-to-neutrophil ratio; PC—plasma cell; HR—hormone receptor; CCL—chemokine C-C motif ligand; CTL—cytotoxic T lymphocyte; PD-L1—programmed cell death ligand 1; RFS—recurrence-free survival; PD-1—programmed cell death 1; CFR—CD8/FOXP3 ratio; B/P—B/plasma cells; T/NK—T/natural killer cells; M/D—myeloid/dendritic cells; AQUA—auto-quantitative analysis; BCSC—breast cancer stem cell; EMT—epithelial–mesenchymal transition.

**Table 5 ijms-26-09959-t005:** Overview of predictive and prognostic factors for NAC in different breast cancer subtypes.

Subtype	Predictive Factors	Prognostic Factors
TN	➢ TILs ➢ Low NLR/high TIL density ➢ Systemic immune mediators ➢ High TIL levels/PD-L1(anticancer drug + ICI) ➢ CD8/FOXP3 ratio ➢ Change in TIL levels ➢ Patient aged <40 years ➢ Higher TCR clonality, PD-L1, higher CD3/CD68 ratios, closer T/tumor cell proximity; T cell proximity ➢ Decrease in TIL levels and PD-L1; TIM-3+ in non-pCR ➢ HRD ➢ LPBC ➢ Higher pre-NAC TIL levels, CD8/CD4 ratio, post-NAC CD4 level ➢ Higher apoptosis scores	➢ Increased OS/BCSS: High TIL levels in RD after NAC ➢ Improved OS/BCSS: Low NLR/high TIL levels ➢ Improved survival: Circulating cytokines with TILs ➢ Distant DFS: Increased TIL levels in RD ➢ Poor RFS/OS: High post-NAC TIL levels ➢ Less distant recurrence or longer distant RFS: PLR ➢ Better OS: High TIL levels; Worse OS: PD-L1/TIM3+; EFS: negatively associated with TIL levels ➢ Improved DFS: Decrease in TIL levels ➢ Improved RFS/OS: Higher TIL density, diffuse TIL pattern ➢ Improved RFS/OS: Higher RD TIL levels ➢ Good prognosis: High TIL levels; Local recurrence: decreased TIL levels ➢ Longer DFS: Higher pre-NAC CD3, CD4, CD8, FOXP3 levels, CD4/FOXP3 ratio; Longer OS: Higher pre-NAC CD3 level
HER2	➢ TILs no association with pCR, trend of association for HR ➢ TIL levels ➢ CD8 infiltration ➢ iICA (Inferred immune cell activity) ➢ Multiple B cell-related signature ➢ sTILs; CD3+, CD3+CD8−FOXP3−, CD8+, FOXP3 sTILs, immune cell aggregates ➢ High TIL/PD-L1+ TIL levels ➢ High TIL (not sTIL) levels, higher sTIL (not TIL) levels during treatment, increased infiltrative lymphocytes in non-pCR, decreased CD8 in pCR ➢ High Ki-67 levels ➢ High CD4+, CD8+, CD20+ sTILs, CD20+ iTIL levels ➢ Decrease in magnitude of TIL level; High post-NAC TIL levels in RD: aggressive disease ➢ LPBC	➢ High TIL scores: Excellent IDFS (3-year), regardless of pCR ➢ Poor prognosis: higher RD TIL levels ➢ Better DFS/OS: iICA cluster ➢ EFS: Multiple B cell-related signature ➢ Improved BCSS/DFS: Increased post-NAC TIL levels ➢ Improved EFS: Increased baseline TIL levels ➢ Good prognosis: High TIL levels; Local recurrence: Decreased TIL levels ➢ Worse prognosis: High post-NAC TIL levels in RD ➢ TILs: More relevant prognostic factor in TPBC
Luminal	➢ TIL levels, immune-related gene signatures, immune cell subpopulations ➢ Gene expression for FOXP3/CD163 ➢ iCD8+ TIL levels	➢ Poor prognosis: TILs > 10%, LMR < 5.2 ➢ Improved DFS: Low FOXP3 levels ➢ DFS/OS: CD8+ TIL levels; OS: iCD8+ TILs
NS	➢ Titin ➢ Pre-NAC sTILs levels, regardless of subtype ➢ High TIL levels, NLR, PLR ➢ CD8/eTreg ratio ➢ PC/B cell infiltration ➢ High TIL levels for TN, HER2, not luminal ➢ Pre-NAC TIL levels for luminal, TN, not HER2; Decreased mean TIL levels after NAC ➢ Higher baseline TIL, chemoattractant cytokine (CCL21, CCL19), CTL marker levels; Deceased TIL levels and most immune gene expression; PD-L1 rate; decreased TIL levels after NAC ➢ Higher TIL levels in TN, tended to be associated with luminal ➢ iTILs levels in luminal; iTIL and sTIL in TN; neither in HER2 ➢ PD-L1/PD-1, increased TIL levels ➢ High TIL levels in TN; CD8 infiltration ➢ iTIL/sTIL levels in HER2/TN, no biomarker in luminal ➢ High CD8/FOXP3 ratios in TN/HER2 ➢ iTIL/sTIL/PD-L1 ➢ B cell/plasma cells, myeloid/dendritic cells, T cell/natural killer cells ➢ TIL levels, stromal AQUA scores for CD3, CD8, CD20	➢ DFS: Lower post-NAC TIL levels and pCR ➢ Improved DFS/OS: pCR for TN/HER2, not luminal ➢ DFS: Pre-NAC TIL levels ➢ DFS: TIL levels ➢ Longer DFS: PC infiltration in HR− ➢ DFS/OS: High TIL levels in TN/HER2; Worse survival for luminal ➢ DFS: Pre-NAC TIL levels; Impaired DFS: High post-NAC TIL levels in HER2 ➢ Improved OS: Increased TIL levels in TN; Shorter OS in luminal ➢ Better RFS: Low post-NAC TIL levels in luminal ➢ Poor prognosis: PD-L1/PD-1 ➢ Longer DFS/OS: High TIL levels ➢ Good prognosis: High CD8/FOXP3 ratio

NAC—neoadjuvant chemotherapy; TN—triple-negative; TIL—tumor-infiltrating lymphocyte; NLR—neutrophil-to-lymphocyte ratio; PD-L1—programmed cell death ligand 1; ICI—immune checkpoint inhibitor; CD—cluster of differentiation; FOXP3—forkhead box P3; TCR—T cell receptor; pCR—pathological complete response; HRD—homologous recombination deficiency; LPBC—lymphocyte-predominant breast cancer; OS—overall survival; BCSS—breast cancer-specific survival; RD—residual disease; DFS—disease-free survival; RFS—relapse-free survival; PLR—platelet-to-lymphocyte ratio; TIM-3—T cell immunoglobulin mucin-3; EFS—event-free survival; HER2—human epidermal growth factor receptor 2; HR—hormone receptor; IDFS—invasive DFS; sTIL—stromal tumor-infiltrating lymphocyte; iTIL—intratumoral tumor-infiltrating lymphocyte; TPBC—triple-positive breast cancer; iCD8—intratumoral CD8; LMR—lymphocyte-to-monocyte ratio; NS—non-subtype-specified; eTreg—effector regulatory T cells; PC—plasma cells; CCL—chemokine C-C motif ligand; CTL—cytotoxic T lymphocyte; AQUA—auto-quantitative analysis; PD-1—programmed cell death 1.

## Data Availability

No new data were created or analyzed in this study.

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
