# Peer review of "Roles of Tumor-Infiltrating Lymphocytes and Antitumor Immune Responses as Predictive and Prognostic Markers in Patients with Breast Cancer Receiving Neoadjuvant Chemotherapy"

_ijms, 2025, doi:10.3390/ijms26209959_

Round 1
Reviewer 1 Report
Comments and Suggestions for Authors
Summary of the Study
This manuscript is a comprehensive review discussing the roles of tumor-infiltrating lymphocytes (TILs) and antitumor immune responses as predictive and prognostic biomarkers in breast cancer patients receiving neoadjuvant chemotherapy (NAC). The review integrates findings from multiple studies addressing different breast cancer subtypes (triple-negative, HER2-positive, luminal, and non-specified), focusing on the relationship between TILs, pathological complete response (pCR), and survival outcomes. The authors summarize molecular mechanisms, TIL functional diversity, and immune-modulating effects of NAC, while also highlighting areas of conflicting evidence across tumor subtypes.
General Comments
The manuscript is timely and addresses an important topic at the interface of tumor immunology and breast cancer therapy. It consolidates a significant body of literature and provides a useful resource for clinicians and researchers. However, there are several issues that should be addressed to improve clarity, organization, and clinical impact:
1. The manuscript is lengthy and somewhat repetitive. A clearer organization and structured summaries per breast cancer subtype would improve readability.
2. The review lacks a critical synthesis of conflicting findings, particularly regarding predictive and prognostic roles of TILs across different subtypes.
3. Clinical applicability is underdeveloped — more discussion is needed on how findings can inform NAC decision-making and patient stratification.
4. Figures and tables should be reorganized to better summarize predictive markers, molecular mechanisms, and clinical outcomes.
Specific Comments
Major Issues
- The review includes extensive information but lacks a concise comparative summary of TIL-related predictive/prognostic roles across triple-negative, HER2-positive, luminal, and non-specified breast cancer. Adding a table summarizing subtype-specific conclusions would make the review more accessible.
- While immune pathways such as immunogenic cell death and PD-L1/PD-1 signaling are mentioned, the mechanistic explanations are fragmented. A dedicated figure showing how NAC modulates immune responses and TIL activity across tumor subtypes would enhance clarity.
- The manuscript cites contradictory results for TILs predicting pCR or survival, but does not analyze the potential reasons (e.g., variability in NAC regimens, differences in TIL quantification methods, tumor heterogeneity). A discussion summarizing these sources of heterogeneity is needed.
- The manuscript should better integrate how these findings could influence NAC decision-making, biomarker-driven therapy selection, and future clinical trial designs.
- Several figures lack sufficient annotations, and the legends do not clearly indicate how data were derived. Summarizing predictive biomarkers, survival associations, and immune pathways in a more structured graphical format is recommended.
Author Response
Reviewer 1
Thank you for your valuable comments and suggestions to improve the clarity of the manuscript. Our responses to your points are as follows.
General Comments
- The manuscript is lengthy and somewhat repetitive. A clearer organization and structured summaries per breast cancer subtype would improve readability.
Response: Based on your suggestion, we have reorganized Sections 4 and 5 as “TILs and Immune Responses as Predictors of Treatment Response and Prognostic Markers,” summarizing them by breast cancer subtype (L126–L236). The section “Classification and Functional Roles of TIL Types” has been deleted due to redundancy.
- The review lacks a critical synthesis of conflicting findings, particularly regarding predictive and prognostic roles of TILs across different subtypes.
Response: To clarify the results regarding the predictive and prognostic roles of TILs across different subtypes, additional explanations have been added to the main text (L304–L306; L339–L345; L352–L358). Conflicting results across different subtypes are summarized in Figure 5.
- Clinical applicability is underdeveloped — more discussion is needed on how findings can inform NAC decision-making and patient stratification.
Response: Clinical applicability in TIL assessment has been added to the main text (L561–L570).
- Figures and tables should be reorganized to better summarize predictive markers, molecular mechanisms, and clinical outcomes.
Response: The presented table is arranged chronologically from the most recent year, compiling essential information focused on subtypes, TILs/other markers, immune phenotypes/genes, protein expression, predictive factors, and prognostic factors. The figures schematically illustrate this information to aid understanding. The text provides detailed annotations and explanations. This is highly valuable for understanding immune activation and clinical outcomes across subtypes, suggesting the complexity of immune-tumor interactions in breast cancer patients receiving NAC therapy. Figure 5 was added to clarify conflicting results. Table 5 was added to summarize predictive and prognostic factors across different subtypes.
Specific comments
Major Issues
- The review includes extensive information but lacks a concise comparative summary of TIL-related predictive/prognostic roles across triple-negative, HER2-positive, luminal, and non-specified breast cancer. Adding a table summarizing subtype-specific conclusions would make the review more accessible.
Response: Additional descriptions regarding predictive and prognostic markers across different subtypes were added (L239–L275). Additionally, Table 5 summarizes the predictive and prognostic factors across different subtypes. Although these results were complex and not universally applicable due to the use of different immune markers and methodologies for evaluating NAC-induced immune responses across studies, they may be useful for understanding the complexity of NAC-induced immune responses in different breast cancer subtypes.
- While immune pathways such as immunogenic cell death and PD-L1/PD-1 signaling are mentioned, the mechanistic explanations are fragmented. A dedicated figure showing how NAC modulates immune responses and TIL activity across tumor subtypes would enhance clarity.
Response: The molecular mechanism by which NAC induces immune activation and enhances anticancer effects is a widely recognized phenomenon. However, the molecular mechanism underlying NAC-induced immune activation may vary depending on the subtype and treatment regimen (different taxanes, HER2-targeted drugs, immune checkpoint inhibitors, etc.). While numerous studies have reported fragmented evidence of NAC-induced immune activation, these include conflicting results, and the detailed mechanisms of NAC-induced immune responses remain incompletely understood. Future research is needed to elucidate the key factors and mechanisms determining immune activation or suppression within the tumor microenvironment.
- The manuscript cites contradictory results for TILs predicting pCR or survival, but does not analyze the potential reasons (e.g., variability in NAC regimens, differences in TIL quantification methods, tumor heterogeneity). A discussion summarizing these sources of heterogeneity is needed.
Response: This study did not specifically evaluate or document differences in NAC regimens, methods for quantifying TILs, or tumor heterogeneity. Instead, they focused on local and systemic immune responses following NAC, assessing TILs and other markers, as well as immune phenotypes/gene and protein expression across different subtypes. While NAC regimens are generally standardized, the use of different taxanes, HER2-targeted therapy, and ICIs in tumors with heterogeneity suggest the potential to induce distinct immune responses within the TME. These possibilities have been noted in previous reports. Descriptions of treatment regimens and TIL quantification methods were added to explain the conflicting results (L284-L292). However, the inconsistent pCR results across different subtypes stem from statistically insufficient sample sizes. More importantly, the imbalance between immune-active and suppressor cells in the TME after NAC is responsible for the conflicting results regarding the prognosis of TN and HER2 subtypes. These possibilities are described in the section “Potential Causes of Contradictory Results,” and further research is needed to clarify changes in immune responses within the TME before, during, and after NAC. TIL quantification was performed using immunohistochemical staining for different immune phenotypes and/or gene expression analysis of immune cells. IHC staining included single-plex or multiplex techniques, and different techniques may introduce bias. TIL assessment was performed according to the International TIL Working Group guidelines, but it depends on the pathologist's skill and tumor heterogeneity, which may introduce bias. These aspects are detailed in the section “Evaluation of TILs.”
- The manuscript should better integrate how these findings could influence NAC decision-making, biomarker-driven therapy selection, and future clinical trial designs.
Response: Additional descriptions regarding the clinical significance of NAC decision-making, biomarker-based therapy, and clinical trials have been added to the main text (L561–L570).
- Several figures lack sufficient annotations, and the legends do not clearly indicate how data were derived. Summarizing predictive biomarkers, survival associations, and immune pathways in a more structured graphical format is recommended.
Response: The methods for obtaining the data are described in the text and tables, the results are presented schematically, and the legend is explained based on the data across different subtypes. These figures are not intended to describe the mechanisms of NAC-induced immune responses affecting tumor shrinkage and prognosis. Further research is needed to elucidate how the immune response is activated or fails after NAC in breast cancer patients, in relation to tumor shrinkage and clinical outcomes.
Reviewer 2 Report
Comments and Suggestions for Authors
This is a detailed review on a complex topic. The concepts discussed are currently thought to be important and the authors do a great job at introducing many concepts and players. The review is very comprehensive and illustrates the role for TIL very nicely.
A few recommendations:
- CRT is mentioned a few times but no references are associated with it. Please add references even your own studies.
- Define and explain significance of "P2RY2 and P2RX7 receptors"
- All tables are very complex. It might need some more explanations
- Create a table to describe predictive markers for each breast cancer type and see if any overlap
The rest of this review is very good and nicely summarizes TIL
Author Response
Reviewer 2
Thank you for your valuable feedback on the manuscript. Our responses to your points are as follows.
- CRT is mentioned a few times but no references are associated with it. Please add references even your own studies.
Response: References 16 and 17 describe in detail the role of CRT in immunogenic cell death during immune activation. These references have been added (L80).
- Define and explain significance of "P2RY2 and P2RX7 receptors"
Response: The statement that “P2RY2 and P2RX7 receptors play an important role in immune activation” was added (L92–L96).
- All tables are very complex. It might need some more explanations
Response: Additional explanations regarding the table are provided (L239–L275).
- Create a table to describe predictive markers for each breast cancer type and see if any overlap
Response: Table 5 was added as an overview of predictive and prognostic factors for NAC in different breast cancer subtypes, describing the predictive markers for each breast cancer type (L271–L275).